# Study on the Interaction Mechanism of Methoxy Polyethylene Glycol Maleimide with Sweet Potato β-Amylase

**DOI:** 10.3390/molecules28052188

**Published:** 2023-02-27

**Authors:** Xinhong Liang, Yaxin Kong, Huadi Sun, Ruixiang Zhao, Lingxia Jiao, Wanli Zhang, Bing Liu

**Affiliations:** 1Henan Institute of Science and Technology, School of Food Science, Xinxiang 453003, China; 2Xinxiang Institute of Engineering, School of Food Engineering, Xinxiang 453003, China

**Keywords:** sweet potato β-amylase (SPA), methoxy polyethylene glycol maleimide, circular dichroism, isothermal titration calorimetry, fluorescence spectroscopy, interaction mechanism

## Abstract

In this study, sweet potato β-amylase (SPA) was modified by methoxy polyethylene glycol maleimide (molecular weight 5000, Mal-mPEG5000) to obtain the Mal-mPEG5000-SPA modified β-amylase and the interaction mechanism between SPA and Mal-mPEG5000 was investigated. the changes in the functional groups of different amide bands and modifications in the secondary structure of enzyme protein were analyzed using infrared spectroscopy and circular dichroism spectroscopy. The addition of Mal-mPEG5000 transformed the random curl in the SPA secondary structure into a helix structure, forming a folded structure. The Mal-mPEG5000 improved the thermal stability of SPA and protected the structure of the protein from breaking by the surrounding. The thermodynamic analysis further implied that the intermolecular forces between SPA and Mal-mPEG5000 were hydrophobic interactions and hydrogen bonds due to the positive values of Δ*H*^θ^ and Δ*S*^θ^. Furthermore, the calorie titration data showed that the binding stoichiometry for the complexation of Mal-mPEG5000 to SPA was 1.26, and the binding constant was 1.256 × 10^7^ mol/L. The binding reaction resulted from negative enthalpy, indicating that the interaction of SPA and Mal-mPEG5000 was induced by the van der Waals force and hydrogen bonding. The UV results showed the formation of non-luminescent material during the interaction, the Fluorescence results confirmed that the mechanism between SPA and Mal-mPEG5000 was static quenching. According to the fluorescence quenching measurement, the binding constant (K_A_) values were 4.65 × 10^4^ L·mol^−1^ (298K), 5.56 × 10^4^ L·mol^−1^ (308K), and 6.91 × 10^4^ L·mol^−1^ (318K), respectively.

## 1. Introduction

β-amylase (E.C. 3.2.1.2) is a maltogenic enzyme ubiquitously found in all organisms to facilitate the starch–maltose conversion by catalysing the hydrolysis of 1,4-α-glucosidic linkages. and liberate maltose units from the non-reducing ends [1]. There are several different sources of β-amylase, namely Bacillus, Bacillus cereus, soybean, sweet potato, and barley, and the β-amylase used for maltose production is from barley. Barley β-amylase has a monomeric structure and is highly active in a suitable pH range (optimum pH 4.35) [2]. However, it is unstable above 55 °C. Sweet potato β-amylase (SPA), a type of active macromolecule, can be obtained from sweet potato waste liquid, which is regarded as an industrial by-product during sweet potato starch processing. SPA is easily affected by the environment, limiting its application in the sugar and beer production industries [3]. This shows the poor thermal stability of β-amylase. It is reported that the β-amylase structure consists of a typical (βα)8-barrel nucleus and a long C-terminal loop. The active site is located in the deep pocket of the (βα)8 barrel. The topology of this enzyme is considered to be optimal for recognizing the non-reducing end of the polysaccharide [4]. Some researchers found that the enzymatic activity and thermal stability of β-amylase could be improved by polyethylene glycol (PEG) modification [5]. Chemical modification of enzymes involves changing the molecular structure of enzymes by chemical means, thus affecting the physicochemical properties and changing the catalytic function of enzymes, generally by replacing or deleting a group of enzyme molecules through chemical reactions. At present, common enzyme protein modification groups are amino, carboxyl, and sulfhydryl groups. Methoxy polyethylene glycol is a polyether polymer compound with high water solubility, methoxy at both ends, hydroxyl capped, polar, and able to form hydrogen bonds with water, which is a hydrophilic polymer compound [6]. Therefore, enzyme modification may be a solution to these enzyme problems or even other enzyme limitations.

Polyethylene glycol (PEG), comprising the repeating units of ethylene glycol, is a non-immunogenic biological compound with the characteristics of amphiphilic, non-irritating, non-immunogenic, non-antigenic, non-toxic, etc. [7,8]. According to a report in 1977, proteins improved by PEG are more effective than the unmodified proteins for therapeutic agents [9]. Since then, protein modification technology has developed rapidly. Enzyme modification can reduce or eliminate the antigenicity of the enzyme and improve enzyme stability. So far, 10 types of PEG-modified proteins have passed through the FAD authentication and authorization in the United States [10,11]. Methoxy polyethylene glycol (mPEG) is a derivative of PEG with similar characteristics to PEG. As such, modifying mPEG could improve the biological properties of the enzyme or protein, such as changing the hydrophobicity and charge, increasing the molecular mass of the enzyme protein, and improving the stability and water solubility of the enzyme protein [12,13]. The modification of enzyme protein with mPEG is mainly due to the polymerization degree of methoxy polyethylene glycol itself and the type of binding functional group [14]. In previous studies, mPEG with different functional groups and polymerization degrees, such as Methoxy polyethylene glycol N-hydroxylsuccinimide ester (NHS-mPEG5000, NHS-mPEG20000), Methoxy polyethylene glycol tosylate (Ts-mPEG5000, Ts-mPEG5000, Ts-mPEG10000, Ts-mPEG20000) and Methoxy polyethylene glycol maleimide (Mal-mPEG5000), was used to modify SPA, and Mal-mPEG5000 was screened out as the best modifier. Previous studies have also found that Mal-mPEG5000 modification significantly improved the SPA activity by 24.06% and enhanced heat resistance. Furthermore, the half-life (t_1/2_) of modified β-amylase (Mal-mPEG5000-SPA) was increased by 42.55%, 54.68%, and 62.27% at 50 °C, 55 °C and 60 °C), respectively [5]. Mal-mPEG5000-SPA can be widely used in beer brewing, beverage, dairy, confectionery, bakery food, and other food industries due to its advantages in improving enzymatic hydrolysis efficiency and reducing production costs. Nevertheless, their interaction mechanism has not been reported yet.

Therefore, the present study aimed to explore the interaction between Mal-mPEG5000 and SPA using spectroscopy and thermodynamics. Additionally, the bond strength and the type of force between them were studied by Isothermal titration calorimetry spectroscopy, Fluorescence, and UV–Vis absorption spectroscopy. Meanwhile, the effect of Mal-mPEG5000 on the functional groups and the secondary structure of SPA was investigated by infrared spectroscopy and circular dichroism. Finally, the effect of Mal-mPEG5000 on the thermal stability of SPA was investigated using various scanning calorimetry. Overall, the study results will provide insights into the interaction mechanism of Mal-mPEG5000 with SPA.

## 2. Materials and Methods

### 2.1. Materials

Sweet potatoes of the Xu Shu No. 22 variety were used in this study and were cultivated in Xinxiang City, Henan. They were harvested during the early November harvesting season and then stored at 10–14 °C. The extraction and purification of SPA were performed according to a previously reported method [5]. The molecular weight of the enzyme was 57.9 kDa, and its purity was 95.7%. Mal-mPEG5000 was obtained from Nanocs (New York, NY, USA; purity ≥ 95%). Other reagents were of analytical grade. The water used was distilled water with no fluorescent impurities.

### 2.2. Preparation of the Solutions

The SPA and Mal-mPEG5000 stock solutions were preprocessed by dissolving in PBS buffer (PBS: 0.05M; pH 6.0). The stock concentrations were 125.0 μM for SPA and 4.0 mM for Mal-mPEG5000, respectively.

### 2.3. Enzyme Assay

Enzyme assay was determined by the dinitrosalicylic acid (DNS) method according to Sagu et al. [15] with minor modification. The enzyme assay was performed at 40 °C, pH 5.8, and 1 mg maltose released per hour from 1.1% soluble starch was defined as a unit of enzyme specific activity. In this paper, the enzyme activity was indicated by specific enzyme activity, expressed as U/mg.

### 2.4. Infrared Spectroscopy Measurements

The infrared spectra were recorded on an FTIR spectrometer (Perkin Elmer, Waltham, MA, USA). The enzyme solution in the presence and absence of Mal-mPEG5000 was frozen, dried into a freeze-dried powder and then stored in the desiccator. Exactly 2.5 g of the sample and 250 mg of KBr were evenly ground and pressed into a transparent sheet on the hydraulic mold with a pressure of 5 × 10^7^~10 × 10^7^ Pa. Interferograms were obtained in a range of 650–4000 cm^−1^ with a resolution of 32 cm^−1^ using 32 scans [16].

### 2.5. Circular Dichroism (CD) Measurements

The CD experiments were performed to investigate enzyme conformation with or without Mal-mPEG5000. Exactly 0.4 mL of the 0.1 mg/mL sample was equipped, and the scanning wavelength between 190 and 280 nm was recorded using the J812 CD spectrometer (Jasco, Tokyo, Japan). The CD spectroscopic data were analyzed by the Deconvolution software to calculate the secondary structure of the enzyme [17].

### 2.6. Differential Scanning Calorimetry (DSC) Measurements

The DSC measurements were implemented on a DSC-Q200 (TA Instruments, New Castle, DE, USA). Exactly 1.5 mg of the sample was placed in a solid aluminum pan, and a blank aluminum pan was utilized for reference. The aluminum cap and pan were pressed into the measuring cell tightly using a platen and placed. The measurement was carried out under a nitrogen flow rate of 40 mL/min and a heating rate of 10 °C/min, and the range was 20–200 °C.

### 2.7. Isothermal Titration Calorimetry (ITC) Measurements

The ITC experiments were performed at 298 K using a Nano ITC-200 (TA Instruments, New Castle, DE, USA) controlled by an ITC run software. SPA and Mal-mPEG5000 solutions were prepared in phosphate-citrate buffer at pH 7.0 at a concentration of 0.02 M. All solutions were extensively degassed for 15 min prior to the experiment. The SPA solution was identified as the substrate and 1 mL of this was injected into the sample cell. The Mal-mPEG5000 solution was regarded as the ligand, and 100 μL of it was loaded onto the rotating injector syringe. It should be noted that the rotating injector syringe must be installed on the machine. The automatic controlled rotating syringe consisted of 3 μL of Mal-mPEG5000 solution aliquots injected into the SPA solution at 200 s intervals. The samples were stirred constantly at 250 rpm to ensure thorough mixing. The control experiments were performed by separately injecting the same volume of Mal-mPEG5000 solution into the buffer. The integrated heat modification was analyzed by the Launch NanoAnalyze software [18]. The association constant (*K_a_)*, stoichiometry (N), and the change in enthalpy (Δr*H*_0_) were determined by a standard independent binding model to fit the calorimetric titration data point. The standard molar Gibbs energy (Δr*G*_0_) and standard molar entropic contribution (*T*Δr*S*_0_) were calculated according to Equation (1) below [19]:(1)$$\Delta rG_{{}^{0}}=-RT\ln(K_{a})=\Delta rH_{{}^{0}}-T\Delta rS_{{}^{0}}$$
where *R* is the gas constant (8.314472 J K^−1^ mol^−1^), and T indicates the temperature in Kelvin (298 K).

### 2.8. UV–Vis Absorption Spectroscopy

The absorption measurements of SPA in the presence and absence of Mal-mPEG5000 were recorded using TU-1810 UV–Vis spectroscopy (Beijing Purkinje General Instrument Co., Ltd., Beijing, China) in the range of 200–400 nm with a 1.0 cm quartz cuvette [20]. The concentration of sweet potato β-amylase was fixed at 0.125 mM, whereas the concentration of Mal-mPEG5000 varied from 62.5 to 312.5 μM. The absorbance values of Mal-mPEG5000-SPA mixtures in the concentration range did not exceed 0.05 at the excitation wavelength.

### 2.9. Fluorescence Spectroscopy

The intrinsic fluorescence of SPA was recorded at 298, 308 and 318K upon excitation at 288 nm through a fluorescence spectrophotometer G9800A (Aglient, Santa Clara, CA, USA) [21]. The sweet potato β-amylase concentration was 125.0 μM, whereas the Mal-mPEG5000 concentration varied from 62.5 to 312.5 μM, and the Mal-mPEG5000 concentration varied from 62.5 to 312.5 μM. The Mal-mPEG5000 quenching results were analyzed according to the Stern–Volmer Equation (2).
(2)$$\frac{F_{0}}{F}{=1+K}_{q}\tau_{0}\left[Q\right]{=1+K}_{sv}\left[Q\right]$$
where F_0_ and F are the measure intensities of SPA fluorescence in the presence and absence of Mal-mPEG5000, respectively; which is given by the result of the quenching bimolecular rate constant K_sv_ = K_q_τ_0_, K_q_ stands for the quenching rate constant; τ_0_ refers to the mean protein lifetime in the absence of the quencher (τ_0_ of a biopolymer can be set to 10^−8^ s^−1^, as reported previously); [Q] is the concentration of the quencher, and K_sv_ denotes the Stern-Volmer constant [22].

The number of binding sites (N), and the binding constant (K_A_) can be computed according to Equation (3).
(3)$$\log\left(\frac{F_{0}}{F}-1\right){=\log[K}_{A}]+\mathrm{Nlog}[Q]$$

The N value is a stoichiometric number, and N > 1 indicates at least one independent number of binding sites.

### 2.10. Statistical Analysis

All the experiments were conducted in triplicate. The results were denoted as means ± standard deviation and the data were analyzed using SPSS22.0 statistical software (SPSS Inc., Chicago, IL, USA), and the significance test (*p* < 0.05) was performed using Duncan’s new complex method. All charts were plotted using Origin 8.0 software (OriginLab, Northampton, MA, USA).

## 3. Results and Discussion

### 3.1. Infrared Spectra Analysis

Infrared absorption spectroscopy can analyze the chemical composition of a substance and infer the structure of the substance according to the position and shape of each peak in the spectrum [23,24]. The changes in the SPA functional groups were analyzed in the presence and absence of Mal-mPEG5000 (Figure 1). The vibration of the peptide bond is mostly adopted for the protein structure analysis. The peptide bond induces several amide vibrations that can explore the structure of a protein in different ways. The Amide A band appeared at 3288 cm^−1^ and 3287 cm^−1^, indicating the formation of an association between the N-H stretching vibration and hydrogen bond. In the Amide B bond, SPA and Mal-mPEG5000-SPA showed weak absorption at 3062 cm^−1^, but the peak at 2974 cm^−1^ showed a remarkable blue shift (2974 cm^−1^–2958 cm^−1^). The absorption of high wavenumbers implied that the higher the energy for vibration, the more stable the group is. The Amide I bond (1725–1750 cm^−1^) was the characteristic frequency of lipid carbonyl (C=O). Mal-mPEG5000-SPA showed an absorption peak at 1725 cm^−1^, indicating the reaction between the carbonyl of Mal-mPEG5000 with the enzyme protein. The stretching vibration of the protein polypeptide skeleton, i.e., C-O from 1600 cm^−1^ to 1700 cm^−1^, was a sensitive region of the protein secondary structure change. Additionally, the Mal-mPEG5000-SPA showed an absorption peak at 1649 cm^−1^ instead of SPA (1641 cm^−1^), indicating a change in the Amide I bond of SPA. The Amide II bond (1500–1600 cm^−1^) mainly reflected the C-N stretching and N-H bending vibration of protein, which was an absorption band superimposed by the α-helix, β-fold, and random curl. A blue shift (1536–1584 cm^−1^) was observed in the Amide II bond. The C-O stretching vibration was 1350–1500 cm^−1^. SPA showed the absorption peaks at 1451 cm^−1^ and 1393 cm^−1^, whereas Mal-mPEG5000-SPA showed no absorption peak at 1451 cm^−1^, which was slightly blue-shifted compared with 1404 cm^−1^. Although SPA and Mal-mPEG5000-SPA had C-O, the frequency of the absorption peak was different. The C-O and C-O-C stretching vibrations in the carboxyl group were between 1000 cm^−1^ and 1350 cm^−1^. SPA showed weak absorption peaks at 1298 cm^−1^, 1244 cm^−1^, 1171 cm^−1^, 1103 cm^−1^, and 1044 cm^−1^, whereas Mal-mPEG5000-SPA showed a weak absorption peak at 1245 cm^−1^ and a strong absorption at 1076 cm^−1^. A comparison of the infrared spectrum of SPA and Mal-mPEG5000-SPA revealed the partial movement of the region, indicating that the group in the SPA was more stable after modification with Mal-mPEG5000.

### 3.2. Circular Dichroism Analysis

Circular dichroism (CD) can detect the secondary structure of protein sensitively [25]. The spiral structure, folded structure, and random coil change in Mal-mPEG5000-SPA and SPA can be obtained via analog computation. The CD spectra of SPA and Mal-mPEG5000-SPA are shown in Figure 2a, showing a negative band at 190–230 nm, which is characteristic of the α-helix in the protein and contributes to the n−π* transfer of the peptide bond in the α-helix [26]. The data were calculated using the Deconvolution program, as depicted in Figure 2b. The result indicated that the secondary structures of SPA and Mal-mPEG5000-SPA were mainly helix structures. Compared with SPA and Mal-mPEG5000-SPA, Mal-mPEG5000-SPA showed an increase in the content of the alpha helix and folded structures and a decrease in the content of random coiling. The alpha helix contributes to the stability of the protein, whereas the random helix confers flexibility in the protein structure, resulting in less stability, where the Mal-mPEG4000-SPA helical structure content was increased by 3%, the folded structure increased by 6%, and the random curl decreased by 9%. The addition of Mal-mPEG5000 transformed the random curl in the SPA secondary structure into a helix and a folded structure. These changes suggest that the addition of Mal-mPEG5000 could make the secondary structure of SPA more stable, which was consistent with the infrared absorption spectroscopy results.

### 3.3. DSC Result

DSC is often adopted to explore the degeneration behavior of proteins and heat changes [27]. The thermodynamic properties of SPA and Mal-mPEG5000-SPA were determined by DSC under the same conditions. The statistical analysis results and thermal characteristic parameters of these samples are listed in Table 1. The initial transition temperature and the final transition temperature (T_0_ and T_C_) of SPA were 21.52 °C and 122.14 °C, respectively. T_0_ and T_C_ of Mal-mPEG5000-SPA were 23.15 °C and 127.90 °C, respectively. The maximum transition temperature (T_m_) of SPA and Mal-mPEG5000-SPA was 70.03 °C and 74.22 °C, respectively. The heat degeneration enthalpy (∆H) was obtained by integrating the peak areas at the maximum temperature. It can be seen from Table 1 that the ΔH of SPA and Mal-mPEG5000-SPA were 118.30 J/g and 149 J/g, respectively. Compared with the native (SPA) and modified enzymes (Mal-mPEG5000-SPA), the T_m_ and H were increased by 4.19 °C and 31.41 J/g, respectively. These results indicate that the heat degeneration of Mal-mPEG5000-SPA needs a higher temperature and more heat than SPA. The Mal-mPEG5000 improved the thermal stability of SPA and protected the breakdown of its protein structure by the surrounding. The IR and CD results were verified further.

### 3.4. Isothermal Titration Calorimetry Investigation

ITC is an important means for thermodynamic characterization of the interaction between Mal-mPEG5000 and SPA. The binding constant, thermal enthalpy, and stoichiometry of the reaction between Mal-mPEG5000 and SPA were calculated, and the thermodynamics of SPA and Mal-mPEG5000 were analyzed based on the precise heat change. Of the binding sites, each peak represented a solo injection of the Mal-mPEG5000 into the SPA solution (Figure 3), and the date points and solid line depicted in Figure 3 best fitted to the experimental results. The data points were fitted with an independent model to compute the *K_a_*, N, and Δ_r_*H*_0_ values. As can be seen from Figure, at 298.15 K, the binding constant was 1.256 × 10^7^, and the N value of the binding reaction was 1.26. The data of K_a_ and N suggests that the reaction between SPA and Mal-mPEG5000 has a strong binding affinity. The value of Δr*H*_0_ and *T*Δ_r_*S*_0_ were −87.29 kJ/mol and −15.78 kJ/mol, respectively. The interaction between SPA and Mal-mPEG5000 shows a relatively strong hydrogen bonding interaction between the two as both Δr*H*_0_ and *T*Δ_r_*S*_0_ values are negative; if both Δr*H*_0_ and *T*Δ_r_*S*_0_ are positive, then hydrophobic forces are indicated [28], thus showing that van der Waals forces and hydrogen bonding are the main interaction forces between SPA and Mal-mPEG5000. Thus, it can be inferred that the binding reaction of SPA was triggered by both negative standard molar enthalpy and entropy contributions. The ITC results show that the SPA and Mal-mPEG5000 have the highest binding affinity.

### 3.5. UV–Vis Absorption Spectra

The overlaid UV–Vis absorption spectra (Figure 4) illustrated the binding of Mal-mPEG5000 and SPA and formed a complex of Mal-mPEG5000 and SPA, which might have happened because the UV absorption intensity of SPA increased and showed a blue shift (281–276 nm) with the increasing Mal-mPEG5000 concentration. SPA is known to be a source of aromatic structure and is therefore, by definition, an aromatic compound capable of participating in the π–π stacking of attractive non-covalent processes [29], by the interceptor molecule hypothesis, explaining specific changes in absorption spectra through heteroconjugation processes interactions. Additionally, the Mal-mPEG5000 peak was subtracted from the Mal-mPEG5000-SPA complex UV–Vis peak (Figure 4), illustrating the change in the protein response upon binding to Mal-mPEG5000 at the wavelength region where native SPA cannot be absorbed i.e., at λ > 310 nm, thereby confirming the formation of a steady-state complex between Mal-mPEG5000 and SPA. Furthermore, a peak such as that found for Trp was observed at 281 nm upon the subtraction of the Mal-mPEG5000 UV spectrum from the mixture. Fluorescence spectroscopy was adopted to explore the interaction between SPA and Mal-mPEG5000 further.

### 3.6. Fluorescence Measurements for Binding Constant and Site

Fluorescence spectroscopy is a highly efficient and reliable tool to explore the binding between different ligands and macromolecules. Fluorescence quenching can provide adequate information on the binding between small protein molecules, including binding mechanism, binding-specific parameters, and structural changes in the protein [30,31]. Fluorescence quenching hypotheses decline the fluorescence spectra intensity of a certain fluorophore using various molecular interactions [32,33]. In this study, Mal-mPEG5000 could linearize the SPA fluorescence quench at different temperatures (Figure 5a–c), and this quenching effect was observed over the emission wavelength range of 288–500 nm after being excited at 288 nm. It can be observed that the fluorescence intensity of enzyme protein decreased in the presence of Mal-mPEG5000 and the maximum emission wavelength shifted from 334 to 339 nm (redshift), indicating a change in the SPA microenvironment after Mal-mPEG5000 addition [34].

The fluorescence intensity quenching of a fluorophore could be induced by both dynamic and static quenching [35]. Molecular diffusion in a solution leads to the dynamic type of quenching, whereas the formation of a ground-state complex refers to the static quenching mechanism. These two quenching mechanisms depend on different temperatures, such as high or low; high temperatures can enhance the quenching constants for the dynamic type of quenching and vice versa [36]. In this study, the Mal-mPEG5000 quenching results were derived from the Stern–Volmer Equation (2). The interaction between Mal-mPEG5000 and SPA produced linear Stern–Volmer plots (Figure 6), illustrating a static mechanism. Furthermore, the Stern–Volmer constants listed in Table 2 declined with increasing temperature, fitting well with the static mechanism. Further confirmation on the steady-state complex formation can be obtained from the values of the quenching rate constants K_q_ (see Table 2) determined from Equation (2). The obtained K_q_ values were greater than the previously reported values in different quenchers with a biopolymer of 2 × 10^10^ L·mol^−1^ s^−1^, indicating it to be a static type of quenching.

The type of interaction force was determined by the change in the thermodynamic parameters of SPA and Mal-mPEG5000. The thermodynamic variables have free energy (Δ*G*^θ^), entropy (Δ*S*^θ^) and enthalpy (Δ*H*^θ^) [37,38]. This binding might be mediated by one or several binding forces, such as hydrogen bonding, hydrophobic, van der Waals and electrostatic forces [39]. In a previous study, the molecular proteins analyzed by fluorescence spectroscopy were as follows: (1) positive Δ*H*^θ^ and Δ*S*^θ^ indicate hydrophobic binding; (2) negative Δ*H*^θ^ and Δ*S*^θ^ refer to the presence of hydrogen bonding or van der Waals forces; and (3) electrostatic forces can be determined by a negative Δ*H*^θ^ and a positive Δ*S*^θ^ [40]. The thermodynamic parameters were calculated as follows:(4)$$\ln K=-\frac{\Delta H^{\theta }}{RT}+\frac{\Delta S^{\theta }}{R}$$
(5)$$\Delta G^{\theta }=\Delta H^{\theta }-T\Delta S^{\theta }$$
where *K* and *R* denote the association and gas constants, respectively, whereas *T* refers to temperature (in Kelvins). Subsequent plotting of ln *K* against 1/*T* (Figure 7) results in the values of Δ*H*^θ^, Δ*G*^θ^ and Δ*S*^θ^ (Table 2).

The combined data of the estimated positive Δ*H*^θ^ and positive Δ*S*^θ^ may also strongly suggest the involvement of hydrophobic interactions and hydrogen bonds. Here, both hydrophobic and hydrogen bond forces were responsible for the interaction between Mal-mPEG5000 and SPA. The negative Δ*G*^θ^ revealed that the binding process was spontaneous, which was consistent with the thermodynamic study of Moringa seed hulls by Lopes [41].

According to Equation (3), the value of n is a stoichiometric number, and N > 1 indicates at least one independent number of the binding sites [42]. The value of N increased as the temperature of the interaction between SPA and Mal-mPEG5000 increased. At 298 K, the N and K_A_ of the fluorescence were 1.27 and 4.65 × 10^4^ L·mol^−1^, respectively (Table 2). By comparing with the isothermal titration calorimetry results, similar n and different K_A_ were obtained. Although isothermal titration calorimetry and fluorescence could measure the interaction between SPA and Mal-mPEG5000, their methods are different; thus, their results would be different also. It is reported that polyphenols could reduce the fluorescence intensity of HAS by a static quenching mechanism. Several studies have reported that the n and K_A_ between ferulic acid and BSA, determined by isothermal titration calorimetry and fluorescence, significantly differ from each other [43].

## 4. Conclusions

In this study, Mal-mPEG5000 treatment successfully influenced the mechanism of SPA. The wave number shift occurred in the infrared spectroscopy, and the random curl in the SPA secondary structure was transformed into a helix and folded structure. The circular dichroism results indicate that the addition of Mal-mPEG5000 leads to a more stable secondary structure of the SPA, which is consistent with the infrared spectroscopy results. The ITC result indicates that SPA and Mal-mPEG5000 had the highest binding affinity. T_0_ further proves that the interaction between SPA and Mal-mPEG5000 could be confirmed by UV and fluorescence spectroscopy. Overall, the study results imply that the addition of Mal-mPEG5000 could make the enzyme protein structure more stable. Compared with the native and modified enzyme, the latter could resist higher denaturation temperatures, indicating that the addition of Mal-mPEG5000 benefited SPA.

## Figures and Tables

**Figure 1 molecules-28-02188-f001:**
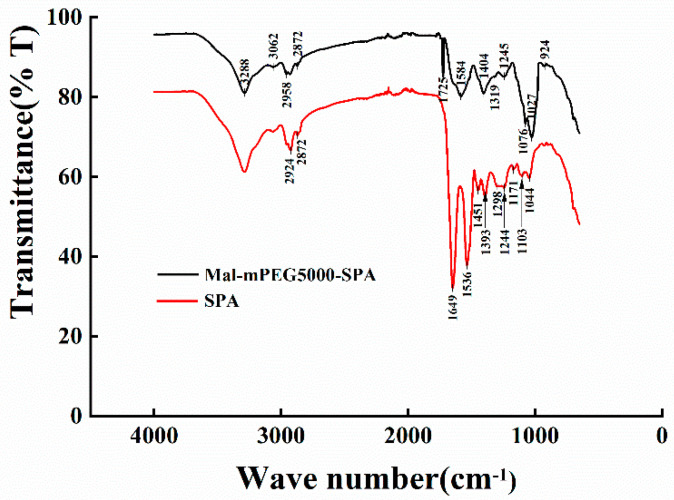
Infrared spectra of SPA and Mal-mPEG5000−SPA.

**Figure 2 molecules-28-02188-f002:**
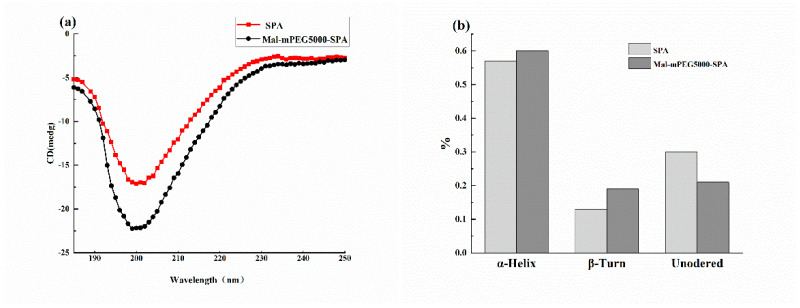
Circular dichromatogram (**a**) and the secondary structure (**b**) of SPA and Mal−mPEG5000−SPA.

**Figure 3 molecules-28-02188-f003:**
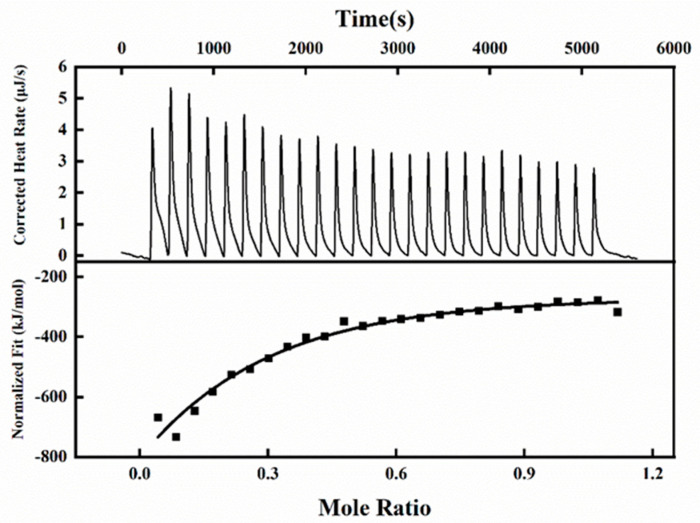
ITC thermograms for the titration of Mal−mPEG5000 with SPA.

**Figure 4 molecules-28-02188-f004:**
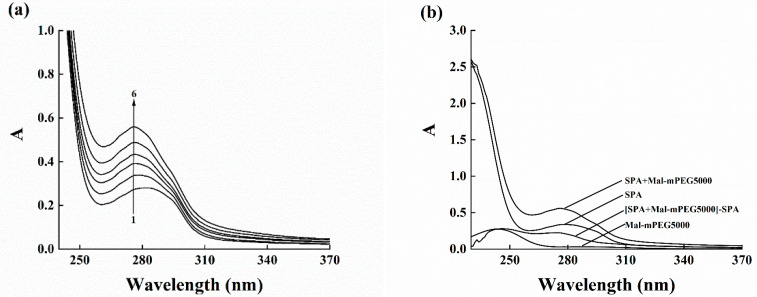
Ultraviolet absorption spectra of Mal-mPEG5000 to SPA. (**a**) UV–visible absorption spectrum of β-amylase upon addition of mPEG at 298 K. c(SPA) = 62.5 μM; c(Mal-mPEG5000) = 0.0, 62.5, 125.0, 187.5, 250.0 and 312.5 μM for curves 1–6. (**b**) UV–visible absorption spectrum of subtraction result.

**Figure 5 molecules-28-02188-f005:**
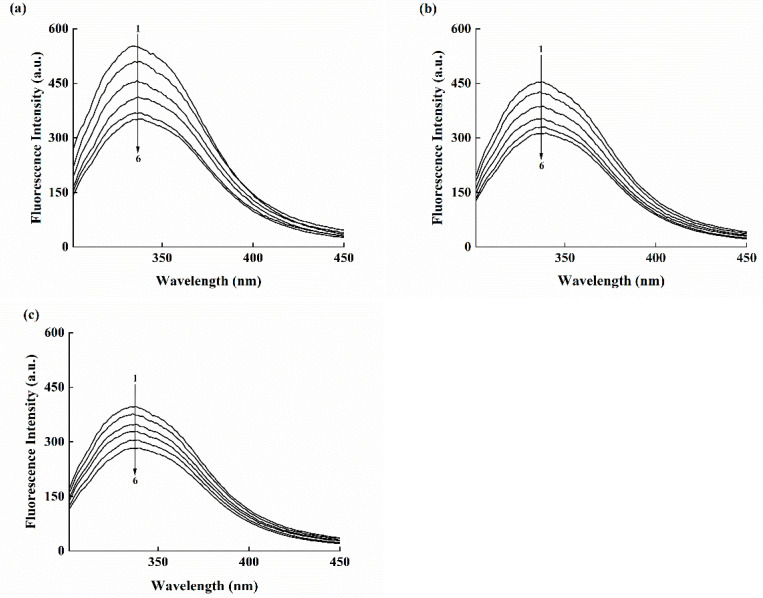
Spectra of the SPA (62.5 μM) fluorescence quenching upon binding to Mal-mPEG5000. c(Mal-mPEG5000) = 0.0, 62.5, 125.0, 187.5, 250.0 and 312.5 μM for curves 1–6, at different temperature (**a**) 298 K (**b**) 308 K (**c**) 318 K.

**Figure 6 molecules-28-02188-f006:**
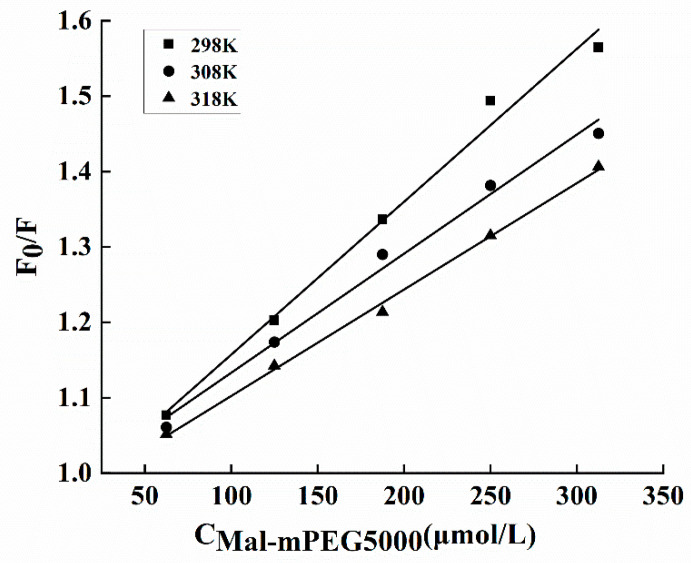
Stern–Volmer plots of SPA interaction with Mal-mPEG5000 at different temperatures.

**Figure 7 molecules-28-02188-f007:**
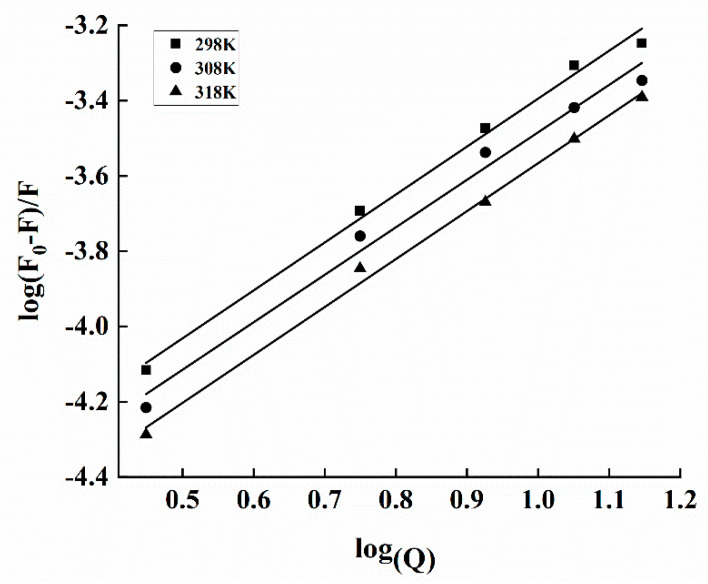
Plots of log[(F0−F)/F] vs. log(Q) for the interaction between Mal−mPEG5000 and SPA at different temperatures.

**Table 1 molecules-28-02188-t001:** Temperature changes in SPA and Mal-mPEG5000-SPA.

Sample	Initial Transition Temperature (°C)	Half-High Transition Temperature (°C)	Final Transition Temperature (°C)	Enthalpy (ΔH) (J/g)
SPA	21.52	70.03	122.14	118.30
Mal-mPEG5000-SPA	23.15	74.22	127.90	149.71

**Table 2 molecules-28-02188-t002:** Summary of the thermodynamic parameters for the interaction between Mal-mPEG5000 and SPA along with binding parameters K and n.

T(K)	K_sv_·10^4^/ (L·mol^−1^)	K_q_·10^12^/ (L·mol^−1^·s^−1^)	K_A_·10^4^/ (L·mol^−1^)	n	Δ*H*^θ^/ (kJ·mol^−1^)	Δ*G*^θ^/ (kJ·mol^−1^)	Δ*S*^θ^/ (J·mol^−1^·K^−1^)
298	4.53 ± 0.14^a^	4.53 ± 0.14^a^	4.65 ± 0.16^c^	1.27 ± 0.1^a^	20.73 ± 0.34	−26.63 ± 0.45^a^	159.14 ± 0.81^a^
308	3.53 ± 0.15^b^	3.53 ± 0.15^b^	5.56 ± 0.21^b^	1.26 ± 0.04^a^		−27.98 ± 0.55^b^	158.16 ± 0.83^a^
318	3.15 ± 0.13^c^	3.15 ± 0.13^c^	6.91 ± 0.18^a^	1.21 ± 0.03^b^		−29.46 ± 0.26^c^	157.83 ± 0.91^a^

* Note: Different superscript lowercase letters in the same row mean significantly different (*p* <  0.05).

## Data Availability

Research data are not shared.

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
