# Peer review of "Study on the Interaction Mechanism of Methoxy Polyethylene Glycol Maleimide with Sweet Potato β-Amylase"

_molecules, 2023, doi:10.3390/molecules28052188_

Round 1

Reviewer 1 Report

The manuscript "Study on the interaction mechanism of Methoxy polyethylene 2 glycol maleimide with sweet potato β-amylase" by Yaxin Kong, Huadi Sunb, Ruixiang Zhaoa, Lingxia Jiaoa, Wanli Zhanga, Bing Liub, and Xinhong Lianga presents interesting new data on enzyme modification.

The authors well formulated the problem, described the experiments, analytical methods and experimental data.

In my opinion, the results obtained in the article are interesting, but they are not discussed in sufficient detail by the authors, taking into account their further possible use, as well as comparison with previously obtained data in this area of research.

Author Response

Dear Reviewer:

Thank you for your letter and for the reviewers’ comments concerning our manuscript entitled “Study on the interaction mechanism of Methoxy polyethylene glycol maleimide with sweet potato β-amylase” (ID: molecules-2235007). We have studied reviewer’s comments carefully and have made revision. We have tried our best to revise our manuscript according to the comments. Attached please find the revised version, which we would like to submit for your kind consideration. We would like to express our great appreciation to you and reviewers for comments on our paper. Looking forward to hearing from you.

Thank you and best regards.

Yours sincerely.

Responds to the reviewer’s comments:

List of changes

Reviewer comment:

In my opinion, the results obtained in the article are interesting, but they are not discussed in sufficient detail by the authors, taking into account their further possible use, as well as comparison with previously obtained data in this area of research.

Response: Thanks for your advices. Relevant content has been added to the manuscript results and discussion section and marked in red font.

Reviewer 2 Report

Thank you very much for offering me a review of the publication. The manuscript titled: “Study on the interaction mechanism of Methoxy polyethylene  glycol maleimide with sweet potato β-amylase” - ID: molecules-2235007 is quite interesting. In this study, sweet potato β-amylase (SPA) was modified by Methoxy polyethylene glycol maleimide to obtain the modified β-amylase Mal-10 mPEG5000-SPA and the interaction mechanism between SPA and Mal-mPEG5000 was investigated.

In my opinion, the manuscript needs to be revised before it is published. First of all, the introduction should be developed, the authors should redraft the chapter "Results and discussion" and the conclusions should be rewritten. The layout of the manuscript is quite chaotic, especially the results and discussion.

GENERAL COMMENTS

Introduction

I think the chapter is pretty short. I propose to thoroughly characterize b-amylases, their structure and importance in the environment. The chapter lacks a precisely articulated research goal and research hypothesis.

Results and Discussion

Subchapter 3.1. “The isothermal titration calorimetry investigation” has not been discussed. Why not?

Subchapter 3.2. “UV–Vis Absorption Spectra” has not been discussed. Why not?

Subchapter 3.5. “Circular dichroism analysis” has not been discussed Why not?

The whole chapter was unfortunately split up. Chaos reigns in this chapter at this stage.

Conclusion

“In this study, the interaction mechanism between Mal-mPEG5000 and SPA was studied using different spectroscopy and thermodynamics.“ - this sentence is not a conclusion.

The changes in the functional groups of different Amide bands and the secondary structure of the enzyme protein were determined and analyzed by Infrared spectroscopy and Circular dichroism Spectroscopy” - this sentence is not a conclusion. What are the research perspectives of these results?

Author Response

Dear Reviewer:

Thank you for your letter and for the reviewers’ comments concerning our manuscript entitled “Study on the interaction mechanism of Methoxy polyethylene glycol maleimide with sweet potato β-amylase” (ID: molecules-2235007). We have studied reviewer’s comments carefully and have made revision. We have tried our best to revise our manuscript according to the comments. Attached please find the revised version, which we would like to submit for your kind consideration. We would like to express our great appreciation to you and reviewers for comments on our paper. Looking forward to hearing from you.

Thank you and best regards.

Yours sincerely.

Responds to the reviewer’s comments:

List of changes

11. Response to comment:

Introduction

I think the chapter is pretty short. I propose to thoroughly characterize b-amylases, their structure and importance in the environment. The chapter lacks a precisely articulated research goal and research hypothesis.

Response: We have made the correction according to the Reviewer’s comments.

The content is increased to lines 31-77 of the manuscript.

2. Response to comment:

Results and Discussion

Subchapter 3.1. “The isothermal titration calorimetry investigation” has not been discussed. Why not?

Response: Thanks for your advices.

The content is increased to lines 374-378 of the manuscript.

Subchapter 3.2. “UV–Vis Absorption Spectra” has not been discussed. Why not?

Response: Thanks for your advices.

The content is increased to lines 398-402 of the manuscript.

Subchapter 3.5. “Circular dichroism analysis” has not been discussed Why not?

Response: Thanks for your advices.

The content is increased to lines 322-325 and 328-333 of the manuscript.

The whole chapter was unfortunately split up. Chaos reigns in this chapter at this stage.

Response: Thanks for your advices.

In the manuscript we have rearranged the structure and combined it in the order of structure and appearance, first determining that Mal-mPEG5000 can bind to SPA and then spectroscopically analysing it.

3. Response to comment:

Conclusion

“In this study, the interaction mechanism between Mal-mPEG5000 and SPA was studied using different spectroscopy and thermodynamics. “- this sentence is not a conclusion.

Response: We have made the correction according to the Reviewer’s comments

Change to "In this study, Mal-mPEG5000 treatment successfully influenced the mechanism of SPA" in the manuscript.

 “The changes in the functional groups of different Amide bands and the secondary structure of the enzyme protein were determined and analyzed by Infrared spectroscopy and Circular dichroism Spectroscopy” - this sentence is not a conclusion. What are the research perspectives of these results?

Response: We have made the correction according to the Reviewer’s comments

Changes made in lines 590-593 of the manuscript

Reviewer 3 Report

This article presents the modification of sweet potato β-amylase by Mal-mPEG5000. The interaction between sweet potato β-amylase and Mal-mPEG5000 is investigated. The characterization of Mal-mPEG5000 modified β-amylase, the effect of Mal-mPEG5000 on the functional groups and the secondary structure of the sweet potato β-amylase, and isothermal titration calorimetry spectroscopy, fluorescence, and UV-Vis absorption spectroscopy are also performed. The experimental work is good, and the figures and data from this research are acceptable, meaning the importance of this work. Before acceptance, I may suggest revisions to this work.

Major points:

1. Keywords “Spectroscopy” and “thermodynamics” may be changed.

2. It is suggested to give the standard definition of β-amylase activity, activity measurement, and enzyme concentration analysis.

3. Does the monomer of sweet potato β-amylase show a molecular weight of 57.9 kDa (line 82)? SDS-PAGE?

4. In section 2.3, what’s the concentration of phosphate-citric acid buffer?

5. It claims that the Mal-mPEG5000 modified sweet potato β-amylase shows improved thermal stability. It is suggested to display the residue activities of sweet potato β-amylase and Mal-mPEG5000 modified sweet potato β-amylase under different temperatures, namely, results of half-life are approved. 

6. Can the Mal-mPEG5000 modified sweet potato β-amylase be analyzed by SDS-PAGE and MS?

7. The writing, language construction, and organization of the text of this submission should be improved carefully. The typescript of this text suffers from several problems. Please check them as well as other expressions carefully throughout this text, and thoroughly assessed for overall construction and grammar. The followings are examples, please check all the expressions throughout this manuscript.

-Line 10, the text “modified β-amylase Mal-mPEG5000-SPA” should be Mal-mPEG5000-SPA modified β-amylase”.

-Line 94, 1mL?

-Line 170, Of the bindingsites?

-Line 338, Mal-mPEG5000? Mal-mPEG5000 modification?

Please revise the English by passing it to a native English speaker or language polishing company.

Author Response

Dear Reviewer:

Thank you for your letter and for the reviewers’ comments concerning our manuscript entitled “Study on the interaction mechanism of Methoxy polyethylene glycol maleimide with sweet potato β-amylase” (ID: molecules-2235007). We have studied reviewer’s comments carefully and have made revision. We have tried our best to revise our manuscript according to the comments. Attached please find the revised version, which we would like to submit for your kind consideration. We would like to express our great appreciation to you and reviewers for comments on our paper. Looking forward to hearing from you.

Thank you and best regards.

Yours sincerely.

Responds to the reviewer’s comments:

List of changes

1. Response to comment:

Keywords “Spectroscopy” and “thermodynamics” may be changed.

Response: We have made the correction according to the Reviewer’s comments.

Keywords changed to "Fluorescence Spectroscopy" " Circular dichroism", and "Isothermal titration calorimetry" in the manuscript.

  1. Response to comment:

It is suggested to give the standard definition of β-amylase activity, activity measurement, and enzyme concentration analysis.

Response: We have made the correction according to the Reviewer’s comments.

The content is increased to lines 154-159 of the manuscript.

  1. Response to comment:

Does the monomer of sweet potato β-amylase show a molecular weight of 57.9 kDa (line 82)? SDS-PAGE?

Response: Thanks for your advices.

The molecular weight was about 55 kDa by SDS-PAGE electrophoresis. Then the protein band was cut off and detected by flight mass spectrometry. It was found that the protein with high similarity in the MASCOT database was β-amylase, with a molecular weight of 57.9kDa.

  1. Response to comment:

In section 2.3, what’s the concentration of phosphate-citric acid buffer?

Response: We have made the correction according to the Reviewer’s comments.

The concentration of phosphate-citric acid buffer was 0.02 M.

  1. Response to comment:

It claims that the Mal-mPEG5000 modified sweet potato β-amylase shows improved thermal stability. It is suggested to display the residue activities of sweet potato β-amylase and Mal-mPEG5000 modified sweet potato β-amylase under different temperatures, namely, results of half-life are approved.

Response: Thank you very much for pointing out this important issue. We agree with your opinion.

The content is increased to lines 98-101 of the manuscript.

  1. Response to comment:

Can the Mal-mPEG5000 modified sweet potato β-amylase be analyzed by SDS-PAGE and MS?

Response: Thank you very much for pointing out this important issue. In our article "Liang, X., Zhang, W., Ran, J., Sun, J., Jiao, L., Feng, L., & Liu, B. (2018). Chemical modification of sweet potato β-amylase by Mal-mPEG to improve its enzymatic characteristics. Molecules, 23(11), 2754. https://doi.org/10.3390/molecules23112754"

we have SDS-PAGE and MS correlation analysis. In this paper, covalent information is given by IR and DSC, unfortunately, the conditions are limited now, please allow us to add relevant information when the conditions are available.

  1. Response to comment:

The writing, language construction, and organization of the text of this submission should be improved carefully. The typescript of this text suffers from several problems. Please check them as well as other expressions carefully throughout this text, and thoroughly assessed for overall construction and grammar. The followings are examples, please check all the expressions throughout this manuscript.

-Line 10, the text “modified β-amylase Mal-mPEG5000-SPA” should be “Mal-mPEG5000-SPA modified β-amylase”.

Response: We have made the correction according to the Reviewer’s comments.

In line 10 of the manuscript has been changed.

-Line 94, 1mL?

Response: We have made the correction according to the Reviewer’s comments.

In line 187 of the manuscript has been changed to 1 mL.

-Line 170, Of the bindingsites?

Response: We have made the correction according to the Reviewer’s comments.

In the manuscript "bindingsites" has been changed to "binding sites".

-Line 338, Mal-mPEG5000? Mal-mPEG5000 modification?

Response: Thanks for your advices.

Line 338 is Mal-mPEG5000. If it is Mal-mPEG5000 modification, the meaning will be repeated.

Please revise the English by passing it to a native English speaker or language polishing company.

Response: Thanks for your advices. We will give English language polishing company to modify.